# Rasch-Validated Italian Scale for Diagnosing Digital Eye Strain: The Computer Vision Syndrome Questionnaire IT©

**DOI:** 10.3390/ijerph19084506

**Published:** 2022-04-08

**Authors:** Natalia Cantó-Sancho, Elena Ronda, Julio Cabrero-García, Stefano Casati, Angela Carta, Stefano Porru, Mar Seguí-Crespo

**Affiliations:** 1Department of Optics, Pharmacology and Anatomy, University of Alicante, 03690 San Vicente del Raspeig, Spain; natalia.canto@ua.es (N.C.-S.); mm.segui@ua.es (M.S.-C.); 2Public Health Research Group, University of Alicante, 03690 San Vicente del Raspeig, Spain; 3Biomedical Research Networking Center for Epidemiology and Public Health (CIBERESP), 28029 Madrid, Spain; 4Department of Nursing, University of Alicante, 03690 San Vicente del Raspeig, Spain; julio.cabrero@ua.es; 5Eye Clinic, Department of Neurosciences, Biomedicine and Movement Sciences, University of Verona, 37129 Verona, Italy; stefano.casati@univr.it; 6Department of Diagnostics and Public Health, University of Verona, 37129 Verona, Italy; angela.carta@univr.it (A.C.); stefano.porru@univr.it (S.P.); 7Mistral–Interuniversity Research Centre ‘Integrated Models of Study for Health Protection and Prevention in Living and Working Environments’, University of Verona, 37134 Verona, Italy

**Keywords:** validation study, asthenopia, surveys and questionnaires, Rasch model, psychometrics

## Abstract

The use of digital devices affects eye health; this can influence the performance of workers. To assess this impact, validated patient-reported outcome questionnaires are needed. The purpose of this study was to validate the psychometric properties of the Italian version of the Computer Vision Syndrome Questionnaire (CVS-Q©) using Rasch analysis. Two hundred and forty-one Italian workers completed an ad hoc questionnaire on anamnesis and exposure to digital devices, and the Italian version of the CVS-Q©. Subsequently, a battery involving three clinical ocular surface and tear tests was performed. The reliability and validity of the scale was assessed using the Andrich Rating Scale Model, and the prevalence of computer vision syndrome (CVS) was calculated. A good fit of both items and persons to the predictions of the Rasch model was observed, with acceptable reliability, unidimensionality, and no or minimal severe differences as a function of gender or age; moreover, good test–retest repeatability, adequate values of sensitivity, reliability, and area under the curve, and adequate construct validity based on clinical tests were obtained. Workers with a questionnaire score ≥ 7 were found to present with CVS. The prevalence of CVS was 76.6%. The CVS-Q IT© is a valid and reliable scale to assess CVS in Italian workers who use digital devices.

## 1. Introduction

Computer vision syndrome (CVS), also known as asthenopia, visual fatigue, or more recently as digital eye strain (DES), is a problem that has been recognised for more than 20 years [1]. It is considered a repetitive stress injury [2], and defined as a set of symptoms, such as dryness, burning, itching, blurred or double vision, tearing, redness, and eye pain, among others [3], that appear due to the prolonged use of digital devices.

It is now considered a major public health issue [4] due to the regular use of digital devices both in the workplace and during leisure time; device use has increased since the beginning of the pandemic caused by SARS-CoV-2 [5], as teleworking or telecommuting has become the best option to maintain safety measures without paralysing work activities [6]. According to European Commission data, between 2009 and 2019, less than 6% of European workers reported working from home on a regular basis, a figure that doubled during 2020. In Italy in particular, this increase was even more significant, from 3.2% to 12.2% [7]. This way of working, which started as a necessity, is becoming the preferred way of working across Europe. However, although it has advantages, such as the possibility of avoiding commutes, greater flexibility in working hours and a better balance between work and social life [8], negative implications on the visual health of workers are expected, such as an increase in visual and ocular symptomatology, which has already been reported in recent studies [5,9,10].

In the assessment of this symptomatology, a problem lies in the imprecise definition of CVS in most published studies, due to the use of unstructured and non-validated questionnaires. This has led to heterogeneous results that have made it difficult to compare this health problem between populations with different characteristics [11,12,13,14]. There are few validated questionnaires for the diagnosis of CVS [15,16], among them, the CVS-Q© (Computer Vision Syndrome Questionnaire) stands out as the most widely used questionnaire, originally designed and validated in Spanish [17]. Its design was based on a review of the scientific literature, and it was validated with a broad consensus among experts from different fields (occupational medicine, epidemiology, preventive medicine, optometry and ophthalmology), by means of a pre-test, a pilot test, and a re-evaluation. It is a self-administered scale that evaluates the frequency and intensity of 16 ocular and visual symptoms related to the use of digital devices. It obtained sensitivity and specificity values above 70%, good test–retest repeatability, and acceptable psychometric properties derived from the Rasch analysis. Since there is a need for questionnaires to be validated in several languages to establish whether there are differences between countries, the research group responsible for the original questionnaire has been developing cultural adaptations for other languages, such as Italian, among others [18,19].

In Italy, studies with unvalidated questionnaires prior to the pandemic report the prevalence of CVS in workers as ranging from 13.3% to 88.6% [20,21,22,23,24,25,26], and there are no known post-2019 studies on CVS in the Italian working population. In most of these studies, ad hoc questionnaires, or a standardised questionnaire developed by the Società Italiana di Medicina del Lavoro ed Igiene Industriale (SIMLII) [27]—which is pending validation—were used [28,29].

Since the availability of a valid and reliable original tool for the measurement of CVS in the working population [17], as well as its translated and adapted version in Italian [18], this study aimed to validate the psychometric properties of the Italian version of the CVS-Q© (CVS-Q IT©). This will allow epidemiological studies to be carried out on how the increasing use of digital devices is affecting the Italian working population.

## 2. Materials and Methods

### 2.1. The Italian Version of the Computer Vision Syndrome Questionnaire (CVS-Q IT©)

The CVS-Q IT© is a self-administered scale that evaluates the frequency and intensity of 16 ocular and visual symptoms related to the use of digital devices. The scoring of the questionnaire follows the procedure of the original version. The 16 items (symptoms) are scored with two rating scales: one for frequency (never, occasionally, often, or always), and one for intensity (moderate, intense). The responses to the two rating scales for each item are combined multiplicatively into a single scale called symptom severity, and the result should be recoded as 0 = 0; 1 or 2 = 1; 4 = 2. Thus, the scoring rule is:(1)$$\mathrm{Score}=\overset{16}{\underset{i=1}{\sum }}{\left(\mathrm{frequency}\;\mathrm{of}\;\mathrm{symptom}\;\mathrm{occurrence}\right)}_{i}\times {\left(\mathrm{intensity}\;\mathrm{of}\;\mathrm{symptom}\right)}_{i}$$

### 2.2. Design, Target Population, and Ethical Aspects

This epidemiological study of Italian workers, with cross-sectional design, recruited participants at the University Hospital of Verona (Italy) from May to July 2019. Workers aged 18–65 years, with Italian as their mother tongue and who were regular users of digital devices during their working day, were included. Workers who wore contact lenses on a daily basis, who were undergoing refractive or cataract surgery, suffering from any ocular pathology, and/or undergoing ocular (including artificial tears) or systemic treatment in the 3 months prior to the study, which could affect CVS symptomatology, were excluded.

The sample size necessary to reliably estimate the Rasch–Andrich rating scale model (RSM) is between 10 and 20 times the number of thresholds to be estimated [30]. Given that our scale had 18 thresholds (16 items and 2 response thresholds), the sample needed to include at least 180 people.

To recruit participants, people attending routine medical health appointments at the Occupational Medicine Unit were offered to participate in the research. All received the participant information sheet detailing the characteristics of the study. Those who agreed to participate signed an informed consent form, and consented to the processing of personal data in accordance with EU Regulation 2016/679. The study was conducted following the standards of Good Clinical Practice and international ethical principles applicable to human research, according to the latest revision of the Declaration of Helsinki. The study was approved by the Ethics Committee of the University of Alicante (UA-2018-02-22) and by the Comitato Etico per la Sperimentazione Clinica delle Province di Verona e Rovigo (41605).

### 2.3. Procedure

First, participants underwent a guided anamnesis, in which sociodemographic information was collected (gender, age, place of work, tasks performed, and department or operational unit to which the worker belonged), general health (if they take any medication, and type and reason for its use), ocular health (eye-related alterations, pharmacological treatment, and ocular surgery), optical correction (use of correction on a regular basis and at work, as well as its design), and exposure to digital devices (daily hours of use of digital devices for work and leisure purposes, years as a digital device worker, work breaks and their duration, and use of air conditioning at work). Based on this information, inclusion and exclusion criteria were applied. The participants included then completed a hard copy of the CVS-Q IT© and underwent a battery of three clinical ocular surface tests in both eyes. The tests were performed from least to most invasive. Tear stability was assessed using Break-Up Time (BUT) as well as the presence of corneal staining using a slit lamp, fluorescein strips, and blue filter. Finally, after 10 min, the tear quantity was evaluated using the Schirmer II test, with an ocular anaesthetic and Schirmer’s absorbent paper strips. For the 3 clinical tests, the normality criteria established by the TFOS DEWS II Report in 2017 [31] were followed. Accordingly, we considered: the BUT to be altered when it was less than or equal to 10 s; the existence of more than 5 staining points to be altered evidence; and the tear quantity to be inadequate when the wet part of the absorbent strip was less than or equal to 10 mm after 5 min. Finally, a sub-sample of 30 participants from the total sample of included workers completed the CVS-Q IT© again after an interval of between 7 and 15 days from the first administration (retest) [32].

An optician–optometrist and a final-year occupational medicine resident collected all information related to the informed consent, the anamnesis, and the CVS-Q IT© in the Occupational Medicine Unit, while the series of clinical tests were carried out by an ophthalmologist and a final-year ophthalmology resident in the Ophthalmology Unit.

### 2.4. Statistical Analysis

#### 2.4.1. Sample Description

A descriptive analysis was performed for all study variables. For categorical variables, the absolute frequency and percentage were calculated. For continuous variables, the mean and standard deviation (SD), both minimum and maximum, were obtained.

#### 2.4.2. Rasch–Andrich Rating Scale Model Analysis

The RSM scale was used to analyse the psychometric properties of the adapted version. The model entails estimating the parameters of the response structure of the rating scale (the response thresholds, i.e., the locations on the trait where the response probability is the same for two adjacent categories) and a general location (difficulty) parameter on the trait for each item.

It was used to examine the following issues:The performance of the rating scale. The two thresholds of the rating scale (a threshold between categories 0 and 1 and a threshold between categories 1 and 2) should advance monotonically, i.e., they should be ordered, and the separation between them should be at least 1.4 logits. In addition, the average measures of the categories should also advance monotonically;The fit of the items and the response structure to the model predictions. For this, infit and outfit mean square error (MNSQ) chi-square statistics were used. Infit MNSQ gives more weight to differences close to the point where item difficulty and subject ability are matched, and outfit MNSQ includes all differences, regardless of the match between difficulty and ability. Ideally, such values should be close to unity, with a critical range of 0.7–1.3, if the fit to the Rasch model is good [33];The fit of persons to model predictions, using the MNSQ statistics referring to persons. This analysis focuses on detecting persons with MNSQ values greater than two and examining their influence on the model parameters;The assumptions of unidimensionality (only one dimension determines the response to the items) and local independence (the response to one item is not influenced by the responses to the other test items once the level in the trait is controlled). Unidimensionality was assessed using the principal component (PC) analyses of Rasch residuals. The variance explained by the first contrast should be <10%, and the eigenvalue of the first contrast should be <1.9 [34]. In addition, the examination of the patterns of item loadings can give information about the relevance of possible secondary dimensions. Local independence is examined using the residual correlations between items: if they are equal to or less than 0.3, local independence can be assumed [35];Measurement error and reliability item–person model. Compared to a global indicator of scale precision, such as the standard error measurement (SEM), IRT models enable us to estimate the information function of the test (and its reciprocal, the standard error function). This function describes the variation of scale precision along with the trait. As a measure of scale reliability in the sample, the Rasch model’s person separation reliability statistic was employed, which is analogous to Cronbach’s alpha (and which we also compute) and uses logits (the linear scores) instead of raw scores. Person separation reliability usually underestimates reliability, whereas Cronbach’s alpha overestimates it. Reliability should be equal to or higher than 0.7 [36];Targeting the difficulty level of the items to the sample. A good alignment between items and persons occurs when a given person’s mean scores are close to 0 logits, which is the value at which the scale is centred and corresponds to the mean of the items. In addition, a joint mapping of item and person locations allows for a more detailed exploration of the target;Analysis of differential item functioning (DIF) and its impact on scale scores. DIF was examined as a function of gender (female vs. male), age (40+ vs. 40−), and version of the questionnaire (Spanish vs. Italian). It was considered an item to have severe DIF if the between-group contrast (DIF size) was >1 and the *t*-Student value was significant at the 0.05 level, after Bonferroni correction (0.05/16 = 0.003). This was followed by an iterative procedure that eliminated a single item at each step to achieve a purified scale without DIF. To examine the impact of DIF on the scale scores, the procedure developed by Tennant was followed [37]. The proportion of estimates that differed by 0.5 logits or more was calculated as an indicator of the impact of DIF on the non-trivial scores.

#### 2.4.3. Test–Retest Reliability

The difference in means between times was tested using the non-parametric Wilcoxon test. The intraclass correlation coefficient (ICC) was calculated based on a mixed effects model with a measure of absolute agreement. The test–retest reliability of the diagnosis of CVS was analysed by calculating Cohen’s Kappa coefficient (k), with its corresponding 95% confidence interval. The acceptable level of ICC and k was set to >0.70 [38].

#### 2.4.4. Criterion Validity—Sensitivity, Specificity and ROC Curve

To assess the criterion validity of the questionnaire, the scale should ideally be compared with a gold standard. Currently, there is no gold standard for measuring CVS, so it was decided to use the same criterion used by the authors of the original questionnaire to define the existence of CVS. The criterion of “occurrence of at least one symptom two or three times a week” obtained after a review of the scientific literature was most widely accepted by different authors [17].

To determine the diagnostic performance of the CVS-Q IT©, the sensitivity and specificity of all possible values of the questionnaire were calculated and the ROC curve was plotted. To find the point that optimised both values, the Youden Index was calculated, which establishes the cut-off point with the highest sensitivity and specificity together. The area under the ROC curve was also calculated, since this can be used to estimate the ability of the questionnaire to diagnose CVS.

#### 2.4.5. Construct Validity Based on Known Groups

To assess the construct validity of the questionnaire, the strategy of comparing two groups, established according to the clinical tests performed (staining, tear quality, and tear quantity), was followed. The differences in the scores obtained in the questionnaire (Student’s *t*-test) and the differences in the prevalence of CVS (Chi-square) between people with altered and unaltered clinical tests were analysed. Although the clinical tests were performed on both eyes, to classify a worker in the group of altered clinical tests, data from a single randomly selected eye were considered and included when at least 2 of the 3 clinical tests performed were failed. To determine which eye to randomly select from each worker, all patient ID numbers were entered into a statistical programme and the numbers were randomized. It was determined that, from the first half, the right eye would be taken, and from the remaining the left eye would be taken.

### 2.5. Prevalence of CVS and Frequency and Intensity of CVS-Q IT© Symptoms

The total prevalence of CVS was also calculated in the sample of included workers, in addition to the frequency of occurrence and intensity with which each of the 16 symptoms that make up the CVS-Q IT© were perceived.

The statistical programmes Winsteps v5.1.5.0 and SPSS version 25 were used to carry out all the analyses.

## 3. Results

### 3.1. Description of the Study Sample

Out of a total of 296 people who agreed to participate, 55 were excluded for different reasons (Table 1). The presence of an ocular pathology at the time of the study was the most frequent reason for exclusion. The most prevalent pathologies were amblyopia, diagnosed dry eye, and retinal pathologies. There were nine people who, after completing the questionnaires, did not attend the Ophthalmology Unit to undergo the clinical tests, and were therefore not included in the study.

The final sample amounted to a total of N = 241 participants. Of these, 64.3% were women and 93.0% used digital devices for 20 h or more per week (Table 2). The mean age of the sample was 45.49 ± 10.96 years (mean ± SD) and the average time spent using digital devices at work was 5.85 ± 1.53 h per day.

### 3.2. Rasch–Andrich Rating Scale Model Analysis

The rating scale thresholds advanced monotonically, with a separation of 3.52 logits, far exceeding the cut-off (Figure 1). The mean scores per category also advanced monotonically: −2.91, −1.45 and −0.25 for categories 0, 1 and 2, respectively. The infit and outfit values of response categories were also acceptable (infit: 1.02, 1.02 and 0.92; outfit: 1.02, 0.94 and 0.89; for response categories 0, 1, and 2, respectively).

The fit of the items to the RSM predictions was good, as indicated by the infit and outfit MNQS values of all items being within the range of acceptability (Table 3). Item locations ranged from −1.09 for feeling that sight was worsening (*sensazione di vedere peggio*) to 1.99 for eye pain (*dolore oculare*). Person fit to the model was also good: only two subjects had MNQS outfit values slightly higher than 2, suggesting that item responses were generally governed by their trait location (symptom severity).

The results of the PC analysis of the Rasch residuals showed the first contrast with an eigenvalue of 2.12, slightly higher than the cut-off to rule out multidimensionality; however, its proportion of variance explained was 9.3, i.e., slightly less than 10%, and thus supported the unidimensionality of the scale. Item loading patterns (three clusters of items) were compatible with the existence of a secondary dimension composed by items 10, 11, 12, and 15, relating to internal symptoms of visual level or image quality. However, this secondary dimension seemed to have little influence, as indicated by both the MNQS values of these items (see Table 3), and the values of the Pearson correlations (disattenuated from the measurement error) between the scores of this cluster, and those of the other two, which were close to 0.8 (the correlation between the other two clusters was 1). As for local dependence, no residual correlation was higher than the cut-off of 0.3. Consequently, the scale seemed to meet the assumptions of unidimensionality and local independence reasonably.

As indicated by the scale’s information function (Figure 2), the highest precision lies in the interval between 0 (raw score = 16) and 1 logit (raw score = 21), with a SEM of 0.48. The person separation reliability (0.72) and Cronbach’s alpha (0.76) values both exceeded the acceptability threshold for reliability.

The mean of the person scores was −2.36 (SD = 0.61); thus, far from 0. The item–person map (Figure 3) revealed that the questionnaire lacked items located at the lower levels of the trait, i.e., lower severity symptom items. These data indicate that the targeting was poor overall, and better at discriminating people with moderate and severe symptoms.

The results of the DIF analysis can be seen in Table 3. In terms of age, only item 12 showed severe DIF, related to the symptom of difficulty in focusing for near vision (*difficoltà nella messa a fuoco da vicino*), indicating higher symptom severity among those under 40 years of age. As for DIF by gender, no items showed severe DIF. Finally, DIF by version was considerably more notable: four items showed severe DIF; however, the impact of DIF on scale scores was small, with only two scores differing by 0.5 logits or more.

A table with the conversion of Rasch scores in logits to raw scores of the CVS-Q IT© questionnaire is included below (Table 4).

### 3.3. Test–Retest Reliability

No differences were observed between the mean scores obtained before and after (*p* = 0.440). A good test–retest repeatability was observed for both scores (ICC = 0.725; 95% CI: 0.496–0.859) and the diagnosis of CVS (k = 0.780; 95% CI: 0.545–1.015).

### 3.4. Criterion Validity—Sensitivity, Specificity and ROC Curves

The highest Youden Index value (Y = 0.631) corresponded to the Rasch score = −2.0700 (CVS-Q IT© score of 7.5 points). Thus, a cut-off point of 7 points could optimise both sensitivity and specificity, which obtained values of 80.0% and 83.1%, respectively. Those workers using digital devices who scored 7 or more points on the questionnaire had CVS. The area under the obtained ROC curve (AUC = 0.874, with a 95% CI: 0.828–0.919 and a *p* ˂ 0.001) indicates that the CVS-Q IT© has a good discriminant capacity (Figure 4).

### 3.5. Construct Validity Based on Known Groups

Following the criteria established to classify people in both groups, it was obtained that, of the 241 participants, 93 (38.6%) had their tests altered and 148 (61.4%) did not. There were statistically significant differences between the CVS-Q IT© scores (*p* = 0.037) in terms of the diagnosis of CVS (*p* = 0.038) between both groups.

### 3.6. Prevalence of CVS and Frequency and Intensity of Symptoms of CVS-Q IT©

The overall prevalence of CVS in the sample was 67.2%. The most prevalent symptoms were blurred vision (*visione sfuocata*), feeling that sight was worsening (*sensazione di vedere peggio*), and headaches (*mal di testa*), with prevalences of 63.5%, 62.3%, and 56%, respectively. The least prevalent were eye pain (*dolore oculare*), coloured halos around objects (*aloni di colori intorno agli oggetti*), and double vision (*visione doppia*), with prevalences of 11.2%, 16.2%, and 17.4%, respectively. All symptoms occurred occasionally and, more frequently, with moderate intensity (Figure 5).

## 4. Discussion

The CVS-Q IT© is the first linguistic, validated version of the original CVS-Q© which can be used in clinical practice for the surveillance of the visual health of workers in Italy. It can also be used to carry out studies to compare results in groups of the adult digital-device-using population with different characteristics. The present study demonstrated that the survey has adequate item and person fit to the model predictions, adequate reliability, unidimensionality, no or minimal severe DIF by gender and age, and adequate construct validity based on ocular surface and tear tests. A worker with a questionnaire score ≥ 7 will have CVS.

In this study, it was decided to carry out parameter estimation using RSM for several reasons. RSM was the model applied in the original version of the questionnaire and, therefore, its use facilitates the comparison of the results. Moreover, RSM is the first choice for questionnaires measuring a single trait (a unidimensional scale) with a set of items sharing an ordered polytomous response scale, such as the CVS-Q© questionnaire. This technique has a high statistical power when using medium sample sizes [39], as in our case, and prevents the characteristics of the instrument from depending on a specific sample; therefore, the estimated parameters in different groups and contexts will be equivalent [40].

When comparing both versions of the CVS-Q© (original vs. Italian), we observed that the Italian version obtained slightly better results in terms of item and person fit to the model, as well as reliability (assessed by means of the person separation reliability). We also observed higher values for sensitivity, specificity, and AUC in the Italian scale (sensitivity = 75% vs. 80%; specificity = 70.2% vs. 83.1%; AUC = 0.826 vs. 0.874) [17], indicating that this scale performs slightly better internally. Additionally, the test–retest analysis revealed good stability over time for the questionnaire, similar to the results of the original CVS-Q©. Finally, the cut-off point of the Italian version was higher than the original questionnaire. Nevertheless, slight changes in the cut-off point are common in different linguistic versions of health questionnaires [41].

The matching of the items to the sample was poor overall, as the mean item severity was significantly lower than the mean symptom severity of the sample. In parallel, the scale had few items measuring mild symptoms, i.e., at the lower level of the trait. Both facts, however, are common in clinical scales [42]. However, the focus of these scales (e.g., CVS-Q©) is precisely for detecting people with moderate or severe symptoms, being uninformative—both in psychometric and substantive terms—at the lower levels of the trait [43]. Consequently, it is a logical feature of a clinical measure rather than a shortcoming of the scale.

With regard to the differential analysis, the fact that item 12, difficulty focusing for near vision (*difficoltà nella messa a fuoco da vicino*), presented severe DIF by age was to be expected, since from the age of 40 years people typically experience a deterioration in their near vision due to presbyopia [44,45]—a phenomenon also reported in the validation of other vision scales [46]. Regarding the DIF (Spanish vs. Italian), there were different items that presented severe DIF in the different versions. The severe DIF of item 1, burning (*bruciore*), could be justified if we take into account that both translators back-translated this item to a conceptually more severe term than the symptom in the original version [18]. Item 2, itching (*prurito*), also presents severe DIF, as it was detected that its back-translation into Spanish might not fit what was intended to be expressed in the original version; so, after consultation with both Italian and Spanish experts in visual health and with the target population (Italian workers using digital devices), it was decided to change this symptom to *pizzicore*. Finally, we found no reason for the severe DIF observed in items 7, eye pain (*dolore oculare*), and 11, double vision (*visione doppia*), which were relatively frequent in the differential analysis [47]. However, the impact of DIF was low, which allows us to assume that the model and the set of item parameters were similar for all comparable groups.

It should also be noted that, in this study, we present the table with the conversion of Rasch scores into logit and raw scores from the CVS-Q IT© questionnaire, as the logit-based scores are more accurate. This is partly because it is a logistic transformation that expands the scale at the upper and lower ends of the range, in relation to the Likert-based score [48], which is useful for users in clinical studies where, for example, small variations of the construct are to be detected, and which require greater precision of the instrument.

Finally, although the main objective of this research was not to determine prevalence, it should be noted that the prevalence obtained in our study (67.2%) was within the range of prevalence estimated before the pandemic in the Italian working population using ad hoc questionnaires, which varied between 64% and 88% [21,24,26]. However, it was higher than the prevalence reported when using the non-validated instrument standardised by SIMLII (between 13% and 51%) [22,23,25]. Nevertheless, it is to be expected that the frequency and intensity of ocular and visual symptoms associated with the intensive use of digital devices will reach increasingly higher values, especially in the working population, partly due to the pandemic and the adoption of telework [6].

Almost the entire sample studied (93.0%) were considered video display terminal workers according to Italian regulations, using digital devices to work ≥ 20 h per week. Additionally, in the remaining 7.0% not considered by the regulations, the average digital devices usage for work was also high, around 15 h per week. The lack of a control group of people not exposed to digital devices at work might seem a priori to be a limitation of our study, as it prevents us from comparing the results according to greater or lesser exposure. However, we believe that, even if a worker does not use digital devices in the workplace, it is not appropriate to consider him/her as not exposed, given that nowadays the use of these devices outside work can be very high. As a strength, it should be noted that many of the problems that occurred when translating and validating an existing instrument into another language are the same as those that occurred when developing the original instrument [48]; therefore, the participation of two of the authors of the original CVS-Q© facilitated the validation of this version. It should also be noted that criterion and construct validity were analysed in this study, completing the face and content validity assessed during the process of translation and cultural adaptation of this scale [18]. Evidence of all types of validity are relevant because of the widespread use of this scale across a variety of applications, including population monitoring, clinical trials, and outcome research.

## 5. Conclusions

The CVS-Q IT© is the first linguistic, validated version of the original CVS-Q©. Our findings indicate that the CVS-Q IT© is a simple, valid, and reliable scale for the assessment and diagnosis of CVS in the adult digital-device-using population in all types of studies. This provides clinicians and researchers with a suitable tool that will help epidemiological registries, decision making for the implementation of preventive measures, interventions, and treatments, and particularly the comparison between exposed populations in different countries (Appendix A).

## Figures and Tables

**Figure 1 ijerph-19-04506-f001:**
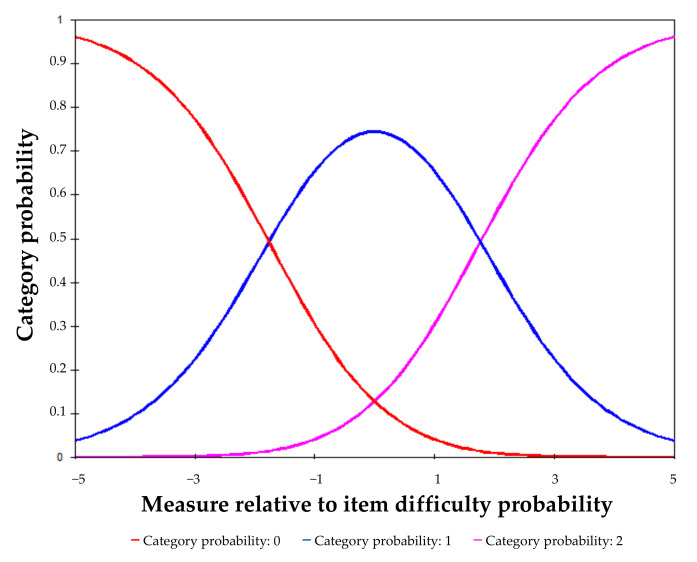
Category response probability curves for the CVS-Q IT©.

**Figure 2 ijerph-19-04506-f002:**
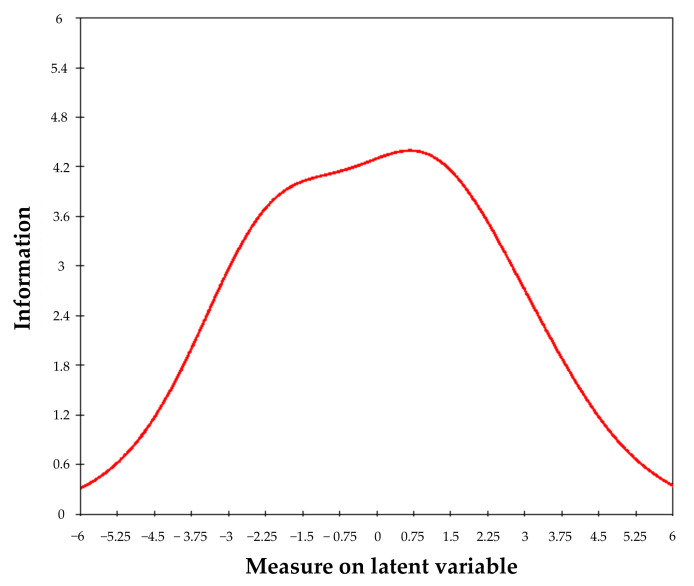
Information function of the CVS-Q IT©.

**Figure 3 ijerph-19-04506-f003:**
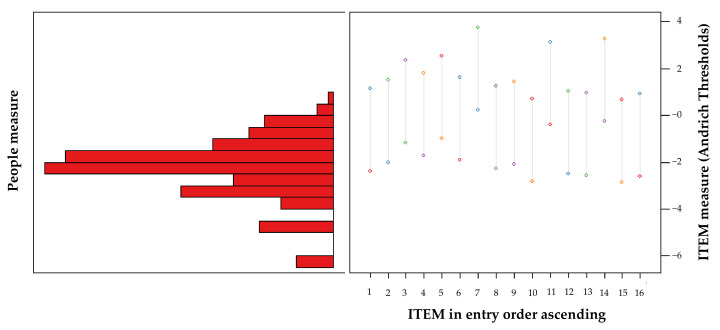
Item–person map for the CVS-Q IT©.

**Figure 4 ijerph-19-04506-f004:**
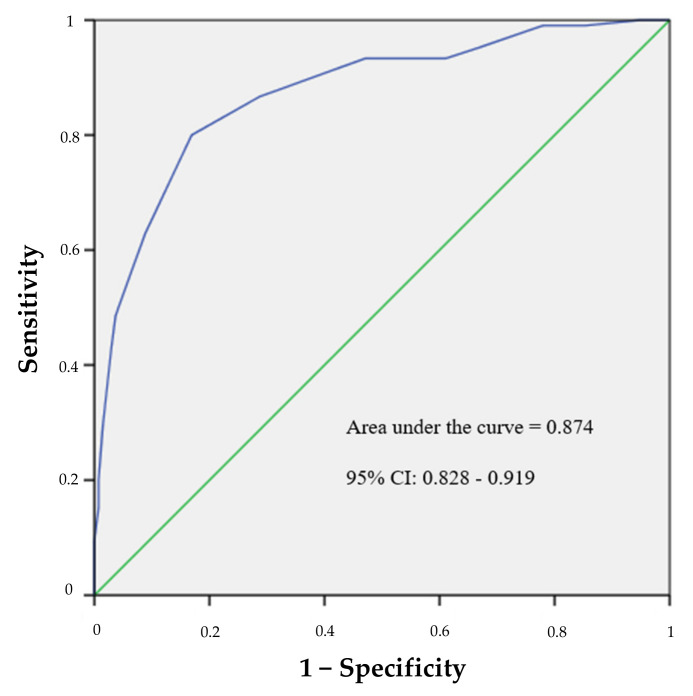
Receiver operator characteristic curve of the CVS-Q IT©.

**Figure 5 ijerph-19-04506-f005:**
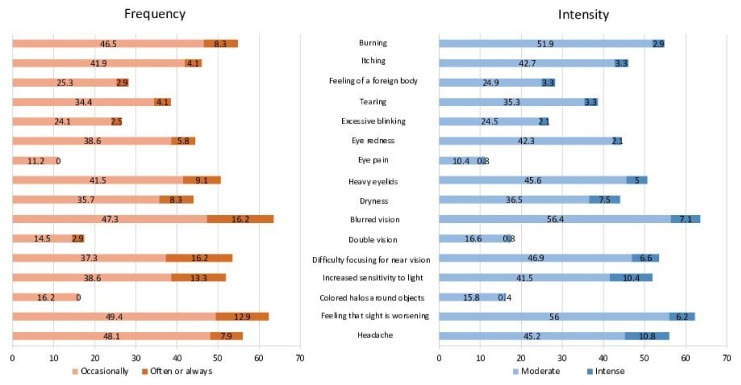
Percentage of workers who presented each symptom according to frequency and intensity.

**Table 1 ijerph-19-04506-t001:** Reasons for exclusion of the participants.

Reason for Exclusion	N
Daily use of contact lenses	6
Prior refractive surgery	8
Prior cataract surgery	4
Ocular pathology at the time of the study	20
Ocular pharmacological treatment at the time of the study	8
Failure to perform the battery of clinical tests	9
Total	55

**Table 2 ijerph-19-04506-t002:** Sociodemographic characteristics and exposure to digital devices of the study sample.

	N	%
Total	241	100
Gender		
Female	155	64.3
Male	86	35.7
Age (years)		
≤35	52	21.6
36–45	55	22.8
≥46	134	55.6
Occupational use of digital devices (hours/week)		
<20	17	7.0
≥20	224	93.0
Years working with digital devices		
≤10	74	30.7
11–20	94	39.0
≥21	73	30.3
Scheduled breaks during work with digital devices		
No	34	14.1
Yes	207	85.9
Use of digital devices to leisure (hours/day)		
<2	130	53.9
≥2	111	46.1
Total use of digital devices (hours/day)		
≤4	9	3.7
5–8	156	64.7
>8	76	31.6

**Table 3 ijerph-19-04506-t003:** Item Rasch analysis results of the symptom severity scale.

Item Description	Severity	SE	Infit MNSQ	Outfit MNSQ	Gender DIF Contrast	Age DIF Contrast	Version DIF Contrast
1. Burning	−0.61	0.13	0.73	0.70	0.12	0.10	* 1.30
2. Itching	−0.24	0.14	0.93	0.94	0.76	0.00	* 1.31
3. Feeling of a foreign body	0.60	0.15	0.93	0.86	0.26	0.29	0.52
4. Tearing	0.06	0.14	1.10	1.10	0.29	0.16	0.25
5. Excessive blinking	0.79	0.16	0.95	0.94	0.47	0.36	0.19
6. Eye redness	−0.15	0.14	1.00	1.06	0.90	0.29	0.67
7. Eye pain	1.99	0.21	1.03	0.82	0.10	0.49	* 1.35
8. Heavy eyelids	−0.49	0.13	0.99	1.01	0.18	0.68	0.72
9. Dryness	−0.33	0.13	1.14	1.05	0.73	0.15	0.08
10. Blurred vision	−1.07	0.13	0.87	0.87	0.03	0.18	0.31
11. Double vision	1.38	0.18	1.02	0.86	0.25	0.05	* 1.41
12. Difficulty focusing for near vision	−0.72	0.13	1.14	1.19	0.20	* 1.58	0.19
13. Increased sensitivity to light	−0.78	0.13	1.25	1.23	0.62	0.15	0.47
14. Coloured halos around objects	1.51	0.19	0.91	0.76	0.29	0.13	0.39
15. Feeling that sight is worsening	−1.09	0.13	0.84	0.81	0.36	0.51	0.20
16. Headache	−0.85	0.13	1.18	1.23	0.32	0.61	0.45

MNSQ, mean square error; DIF, differential item functioning. * *p* ≤ 0.003 (Bonferroni correction).

**Table 4 ijerph-19-04506-t004:** Raw score in the CVS-Q IT© questionnaire and its conversion to logits.

Raw Score	Rasch Score (Logits)	Raw Score	Rasch Score (Logits)
0	−6.05 E	17	0.24
1	−4.78	18	0.47
2	−4.00	19	0.70
3	−3.50	20	0.93
4	−3.11	21	1.16
5	−2.78	22	1.39
6	−2.48	23	1.63
7	−2.20	24	1.88
8	−1.94	25	2.15
9	−1.69	26	2.43
10	−1.44	27	2.75
11	−1.19	28	3.11
12	−0.94	29	3.53
13	−0.70	30	4.07
14	−0.46	31	4.90
15	−0.22	32	6.20 E
16	0.01		

## Data Availability

Not applicable.

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
