# Peer review of "Rasch-Validated Italian Scale for Diagnosing Digital Eye Strain: The Computer Vision Syndrome Questionnaire IT©"

_ijerph, 2022, doi:10.3390/ijerph19084506_

Round 1

Reviewer 1 Report

Well written and presented. Only thing needed is to highlight the novelty in discussion and conclusion sections. 

Reviewer 2 Report

In my opinion, I consider that the manuscript is well designed and necessary, since the computer vision syndrome is becoming more and more frequent.
The only comment is that the authors should structure and explain the methodology and results more clearly, since they are a bit difficult to understand.

Reviewer 3 Report

The manuscript depicts a situation in one old way. in 2k era use of computer screen is so wide that it is impossible avoid it. in this view the paper has low scientific useful significance. with actual computer screen there is a wide spectrum of blue light emission, definition (pixels), refresh time (from 60 to 400Hz) that everyone interested in this topic must be aware. none of these considerations are taken in the report, neither if used devices are registered and correlation with type of the sreen are drawn.

furthermore no considerations are indicated on patients refractive error and near glasses wearing. lacking of this indication is a very important flaw, because this parameter is crucial and it is hard to understand if ocular disconfort syndrome is due to video use or lacking of eye corrections for near.

Reviewer 4 Report

Thank you for the opportunity of reviewing this manuscript (ijerph-1603111). I would like to congratulate the authors for this study on a very important issue. In my opinion this is a piece of research of academic and clinical interest for diagnosing digital eye strain. This paper proves refinement of the Computer Vision Syndrome Questionnaire IT© for Clinical and Educational Settings. I hope that the considerations contributed will help them to improve the manuscript. To facilitate the task, I have some questions about some crucial points:

1) Page 4: IRT application requires two important assumptions: (1) the construct being measured is in fact unidimensional and (2) the items display local independence. Please clarify local independence in table or excel using Winsteps.

2) Page 2: the scale has 18 thresholds (16 items and two response thresholds), Please clarify the scoring method and what Likert points (?) are scored. Page 5: 2.3.5. For construct validity testing, I recommend responses to items in these instruments are ordinal data (the Likert-based score), the authors should apply to Weighted Least Square Mean and Variance adjusted (WLSMV)/Robust Diagonally Weighted Least Squares (RDWLS) to accommodate categorical data. Please use R or Mplus instead of AMOS.

3) Please clarify how to randomize and how to ensure the quality of data collection (the only place: a single randomly selected eye, other?)? Is there any missing data? How to deal with it? How about the sampling strategy? It would be useful to describe it more. More information about the sampling method is needed here. Furthermore, where, when, and how were participants approached? What mode of survey administration was used (online or paper-and-pencil)?

4) Page 4 DIF: age (40+ vs. under 40) should be changed to “(40+ vs. 40-)” or “>40 vs. £40” Why choose 40 as the cutoff value? version of the questionnaire (Spanish vs. Italian), Why did you choose these two languages for DIF?

5) The manuscript should benefit of a final proof editing.

Take care and my best,

Your reviewer

Reviewer 5 Report

This  study demonstrates that the CVS-Q IT© is a valid and reliable scale to assess CVS in Italian workers who use digital devices.

Round 2

Reviewer 2 Report

Accept in the current form

Reviewer 3 Report

The manuscript may be considered for pubblication